# Flow Matching Neural Processes

**Hussen Abu Hamad**
Department of Computer Science
University of Haifa

**Dan Rosenbaum**
Department of Computer Science
University of Haifa

## Abstract

Neural processes (NPs) are a class of models that learn stochastic processes directly from data and can be used for inference, sampling and conditional sampling. We introduce a new NP model based on flow matching, a generative modeling paradigm that has demonstrated strong performance on various data modalities. Following the NP training framework, the model provides amortized predictions of conditional distributions over any arbitrary points in the data. Compared to previous NP models, our model is simple to implement and can be used to sample from conditional distributions using an ODE solver, without requiring auxiliary conditioning methods. In addition, the model provides a controllable tradeoff between accuracy and running time via the number of steps in the ODE solver. We show that our model outperforms previous state-of-the-art neural process methods on various benchmarks including synthetic 1D Gaussian processes data, 2D images, and real-world weather data.

## 1 Introduction

Recent advances in generative machine learning are primarily driven by models capable of leveraging context information to enhance their predictions. To be effective across varying levels of contextual information, these models must handle multiple degrees of conditional uncertainty and therefore accurately capture conditional distributions at multiple levels of abstraction.

The neural process (NP) framework introduced by Garnelo et al. [13, 14] provides a principled approach to context-based predictions by formulating the problem as learning stochastic processes from data, essentially training generative models of functions. Similarly to Gaussian processes, NP models can be used as priors over functions capable of generating samples of arbitrary points along the functions. These target points can be predicted unconditionally or conditioned on a context of observed points along the function. This formulation has made neural processes a popular approach for modeling data in various domains.

In recent years, many different models have been proposed within the neural process framework, differing in their architecture and stochastic mechanisms. Early models focused on stochastic latent variables and were trained by variational inference [14]. More recent approaches such as Nguyen & Grover [32] leverage transformer-based architectures [40] in an autoregressive setup, achieving improved performance in many scenarios.

Despite these advancements, several challenges remain unresolved. With the exception of autoregressive approaches, models tend to underfit the training functions, failing to capture intricate structures. On the other hand, autoregressive models, while more expressive, require sequential sampling of the target points one at a time and are therefore expensive to sample from. Additionally, these models lack a global or hierarchical representation of the function and the uncertainty which can limit their applicability and harm their performance in some cases. For example, the first dimensions in the autoregressive order are constrained to simple distributions.

39th Conference on Neural Information Processing Systems (NeurIPS 2025).

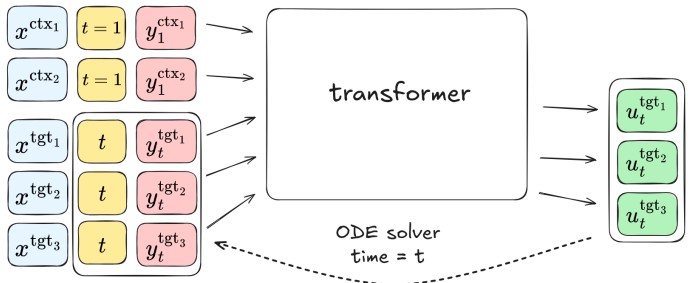

Figure 1: FlowNP: we model a probability flow of a stochastic process where at each step our model takes the values of observed context points from the given function (ctx), and an intermediate value of the target points (tgt) at time t and predicts the velocities of the target points. With an ODE solver this can be used as a model of $p\left(y^{\mathtt{tgt}}|x^{\mathtt{tgt}}, \{x^{\mathtt{ctx}}, y^{\mathtt{ctx}}\}\right)$ to generate samples or compute likelihoods.

In this paper, we introduce FlowNP, a new neural process model based on *flow matching* - a recent generative modeling paradigm that is closely related to diffusion and score-based models and has demonstrated strong performance in images and other modalities [27, 25, 1]. Flow matching operates by continuously transforming a simple and tractable initial distribution into the target data distribution by flowing through a continuous path of intermediate probabilities. The approach enables both generating samples and computing log-likelihood over data by solving an ODE.

Our model is built on a transformer architecture where the input tokens include both the observed context points and intermediate values of the target points as generated across the flow path. At each step, the model outputs a velocity vector which guides the sampler in updating the target point values for the subsequent step. This formulation allows FlowNP to capture complex structures effectively while enabling parallel sampling of all target points.

FlowNP offers several advantages over existing approaches: (1) it is simple to implement, train and use for sampling and inference, (2) it is capable of generating samples and computing likelihoods of any conditional distribution, (3) in contrast to autoregressive models, it generates all target points in parallel, providing a more globally coherent and consistent representation of uncertainty, and (4) inference time is controlled by the number of ODE steps rather than the number of target points. Extensive experimentation on standard NP benchmarks demonstrate that FlowNP consistently outperforms prior NP models, achieving state-of-the-art results across multiple datasets.

Our implementation is available at `https://github.com/danrsm/flowNP`.

## 2 Background

### 2.1 Neural Processes

Neural processes (NP) [13, 14], provide a framework for training generative models of functions, by modeling a *stochastic process* using a dataset of functions. Given such a dataset, a model is trained to predict an arbitrary *target* set of points along a function, based on a *context* set of observed points along the same function. More formally, if the functions are defined as $\mathcal{F} = \{f : \mathcal{X} \to \mathcal{Y}\}$, models are trained to predict a conditional distribution over the target set:

$$p_\theta\left(y^{\mathtt{tgt}}|x^{\mathtt{tgt}}, \{x^{\mathtt{ctx}}, y^{\mathtt{ctx}}\}\right) \tag{1}$$

where $x^{\mathtt{ctx}} \in \mathcal{X}^M$ and $x^{\mathtt{tgt}} \in \mathcal{X}^N$ are the positions of the $M$ context points and $N$ target points respectively, and $y^{\mathtt{ctx}} \in \mathcal{Y}^M$ and $y^{\mathtt{tgt}} \in \mathcal{Y}^N$ are the function evaluations of these points $y_i = f(x_i)$. $\mathcal{X}$ and $\mathcal{Y}$ can be of arbitrary dimensions. The context is placed in '{}' brackets for better readability.

This approach of modeling stochastic processes follows from Kolmogorov's extension theorem [42] stating that stochastic processes can be defined via a collection of joint distributions over finite sets, $p_{x_{1:n}}(y_{1:n})$, if it meets two conditions: exchangeability and consistency.

**Exchangeability.** This condition requires that all joint distributions over the sets are equivariant with respect to permutations:

$$p_{x_{1:n}}(y_{1:n}) = p_{\pi(x_{1:n})}(\pi(y_{1:n})) \coloneqq p_{x_{\pi(1)},\ldots,\pi(n)}(y_{\pi(1),\ldots,\pi(n)}), \tag{2}$$

for any given permutation $\pi$. Models that directly predict conditional distributions such as in Eq. 1 should also be invariant to permutations in the context set.

**Consistency.** This condition requires that joint distributions are consistent with marginalizations:

$$p_{x_{1:m}}(y_{1:m}) = \int p_{x_{1:n}}(y_{1:n})dy_{m+1:n}, \quad \forall 1 \leq m \leq n. \tag{3}$$

For models that directly predict conditional distributions as in Eq. 1, this translates to the following formulation of *marginal consistency*:

$$p_\theta(y_{1:m}|x_{1:m}, \{x^{\text{ctx}}, y^{\text{ctx}}\}) = \int p_\theta(y_{1:n}|x_{1:n}, \{x^{\text{ctx}}, y^{\text{ctx}}\})dy_{m+1:n}, \quad \forall 1 \leq m \leq n, \tag{4}$$

and additionally, models should hold *conditional consistency*, i.e. the chain rule:

$$p_\theta(y_{s^1+s^2}|x_{s^1+s^2}, \{\}) = p_\theta(y_{s^1}|x_{s^1}, \{x_{s^2}, y_{s^2}\}) p_\theta(y_{s^2}|x_{s^2}, \{\}), \tag{5}$$

for any two sets of indexes $s^1$ and $s^2$. We note that while most NP models follow the exchangeability condition, they do not fully guarantee that the consistency conditions hold (see App. A).

## 2.2 Flow Matching

Flow matching [25, 27, 1, 2] is a generative modeling paradigm that has recently achieved significant traction. It is closely related to diffusion modeling and its appeal stems from being both conceptually very simple and versatile.

The fundamental object in flow matching is a probability path $p_t(x)$ defined over a continuous time parameter $t \in [0, 1]$. This path defines a smooth transition between the two distributions $p_0(x)$ and $p_1(x)$. Most commonly, $p_0(x)$ corresponds to a simple tractable distribution such as a Gaussian, and $p_1(x)$ corresponds to a target distribution which is defined only through samples from the training data such as images.

In its simplest formulation, which we also adopt here, training is performed by generating samples of a conditional probability path $p_t(x_t|x_1)$ computed as an interpolation $tx_1 + (1-t)x_0$ between a data sample $x_1$ and a noise sample $x_0$. Given this sample, a model is trained to predict a conditional velocity $u_t(x_t|x_1) = x_1 - x_0$, using a squared loss:

$$\mathcal{L}(\theta) = \mathbb{E}_{t,x_1,x_t} \|u^\theta(x_t, t) - u_t(x_t|x_1)\|^2 \tag{6}$$

It can be shown that the expectation over $x_1$ results in the model approximating the unconditional velocity $u_t(x_t) = \frac{d}{dt}x_t$ defining an ordinary differential equation (ODE) which can be used for both sampling and likelihood estimation.

For sampling, we first generate a sample from the noise $p_0(x_0)$ and then use an ODE solver to transform it into a sample from the data distribution $p_1(x_1)$. To compute the likelihood of a given sample $x_1$, we employ the change of variable formula. This involves transforming $x_1$ back to a corresponding noise sample from the tractable distribution $p_0(x_0)$, and then calculating the likelihood of $x_0$ while correcting for the flow's volume change with an estimate of the Jacobian trace.

## 2.3 Related Work

**Neural processes.** A prominent approach to form distributions over continuous functions is by using Gaussian processes [36]. While they can be powerful, leveraging their capacity in modeling arbitrary functions is limited due to the complexity of training and evaluating them. To this end, neural processes were introduced, allowing the prior over functions to be learned from data in a straightforward way. The first model in the NP class is the Conditional Neural Process (CNP) [13] which is deterministic and predicts an independent Gaussian distribution for each target point given the context. Later, the Neural Process (NP) [14] introduced a latent variable to allow capturing global uncertainty over the functions. Empirical evaluations of the approach was conducted by Le et al. [22]. Following, different extensions to the model were proposed. These include using attention mechanisms (ANP) [21], translation invariance through convolutions [15], bootstrapping [23], and more [8, 18, 3, 4].

Among the extensions of NP, a notable example is the Transformer Neural Process (TNP) [32] which treats the prediction of conditional distributions as sequence modeling. Of the three proposed variants in the TNP paper, the autoregressive model (TNP-A), implemented with a causal mask, achieves the highest scores on most common benchmarks, and here we refer to it simply as TNP. Autoregressive-CNP [7] shows that models can be effectively deployed as autoregressive NPs even if they were not trained autoregressively. However, the autoregressive approach has limitations that we address here, namely, the potential complexity of generating samples point-by-point, and the failure to capture complex distributions in the first points in the autoregressive order. An interesting model that combines autoregressive modeling with diffusion [6] treats the full sampling process as a sequence. Our model is implemented with a transformer, like TNP, however, rather than predicting the next single target point in an autoregressive fashion, we use the transformer as a predictor of the flow matching velocity for all target points at once. We show that our model outperforms the TNP.

**Diffusion on continuous spaces**  Following the success of diffusion models for data in discrete and finite spaces such as image grids [37, 38, 17, 39] different works have investigated their extension to infinite and continuous spaces [19, 34, 35, 41, 5, 30, 20]. These approaches use diffusion or flow matching to model joint distributions with some spatial structure. Given a joint distribution several approaches can be applied to predict conditional distributions. Pidstrigach et al. [35] and Bond-Taylor & Willcocks [5] use a guidance term computed based on the conditioning context and Kerrigan et al. [19] and Dutordoir et al. [11] use a replacement method [29] to generate conditional samples.

Using joint distributions over continuous spaces to predict conditional distributions can be seen as an implemention of neural processes using the definition of conditional distributions $p(x|y) = p(x,y)/p(y)$. The Neural Diffusion Process (NDP) [11] makes this connection explicitly and uses this method to evaluate the model on NP benchmarks. In comparison, our model uses a flow matching formulation, and is trained to capture both joint distributions and arbitrary conditionals directly, thus amortizing the computation of conditioning. We show that our model outperforms the NDP on standard benchmarks, and that amortizing the conditioning enables directly generating conditional samples without needing auxiliary conditioning methods such as guidance or replacement.

## 3   The FlowNP Model

Our goal is to implement a model of conditional distributions defined by any arbitrary target and context sets of points coming from an underlying function as formulated in Eq. 1.

We implement our model using a transformer architecture [40] that predicts the velocities of the target variables at time $t$ in the probability path defined by the continuous flow. The model is depicted in Fig. 1. We perform full self-attention between all input tokens, whereby each of the tokens are updated using an attention operator on all other tokens. The tokens we feed to the transformer are divided to context tokens and target tokens.

**Target tokens:**   These tokens represent the intermediate values of the $N$ function points $y^{\mathrm{tgt}}$ evaluated at the $N$ target positions $x^{\mathrm{tgt}}$. For the evaluation at time $t$ in the probability path, each token is formed by concatenating a single target point position together with the time $t$ and the intermediate value of the variable at that time.

$$\mathtt{token}^{\mathrm{tgt}_i} = \mathtt{embed}([x^{\mathrm{tgt}_i},\ t,\ y_t^{\mathrm{tgt}_i}]) \tag{7}$$

**Context tokens:**   These tokens represent the observed points along the function, on which the distribution is conditioned. They are formed in the same way as the target tokens, except that they always contain the true observed function evaluations $y_1^{\mathrm{ctx}} = y^{\mathrm{ctx}}$, and the time $t = 1$, equivalent to the data distribution $p_1(y)$.

$$\mathtt{token}^{\mathrm{ctx}_i} = \mathtt{embed}([x^{\mathrm{ctx}_i},\ 1,\ y_1^{\mathrm{ctx}_i}]) \tag{8}$$

**Output tokens:**   The output tokens corresponding to the input target tokens are each projected to the original dimension of the function values $\mathtt{dim}(\mathcal{Y})$ and used as the velocity vector at time $t$ in the continuous probability path defined by the model.

## 3.1 Training

We train our model following the standard NP training setup using a conditional flow matching approach. At each training step, a minibatch of functions is extracted from the training set, where for each function two random sets of points are used as the context and target sets. For each step we randomly define different sizes of these two sets. Given the context and target sets, we train our model as a velocity predictor by generating a random sample from an intermediate distribution along the continuous probability path $y_t^{\text{tgt}} \sim p_t(y_t^{\text{tgt}}|y_1^{\text{tgt}})$. This is achieved by interpolating the clean data with random noise:

$$y_t^{\text{tgt}} = ty_1^{\text{tgt}} + (1-t)y_0^{\text{tgt}}, \ y_0^{\text{tgt}} \sim \mathcal{N}(0, I). \tag{9}$$

The loss is computed using a squared error between the model's output and the conditional velocity of the target points $u_t(y_t^{\text{tgt}}|y_1^{\text{tgt}}) = y_1^{\text{tgt}} - y_0^{\text{tgt}}$. Since the model has access to the clean values of the context points, it effectively has side information about the velocity of target variables. In summary, the training loss is computed by:

$$\mathcal{L}(\theta) = \mathbb{E}\|u^\theta(y_t^{\text{tgt}}, t, x^{\text{tgt}}, x^{\text{ctx}}, y_1^{\text{ctx}}) - (y_1^{\text{tgt}} - y_0^{\text{tgt}})\|^2 \tag{10}$$

where the expectation is over:

$$f \sim \mathcal{F}, \{x^{\text{tgt}}, y_1^{\text{tgt}}, x^{\text{ctx}}, y_1^{\text{ctx}}\} \sim f, t \sim \mathcal{U}, y_0^{\text{tgt}} \sim \mathcal{N}$$

The training process is presented in Alg. 1 in the appendix.

## 3.2 Evaluation

Once the model is trained, it can be used both for generating samples and computing likelihoods over data. In order to use the model to generate samples, random values of the target set at time $t = 0$ are drawn from a Normal distribution $y_0^{\text{tgt}} \sim \mathcal{N}(0, I)$ and used as the initial condition when solving the ODE from $t = 0$ to $t = 1$. This can be done with various ODE solvers where each evaluation of the velocity $u_t$ is performed by calling the model with the additional input of target positions and context positions and values, $u_t \approx u^\theta(y_t^{\text{tgt}}, t, x^{\text{tgt}}, x^{\text{ctx}}, y_1^{\text{ctx}})$. The sampling process is presented in Alg. 2 in the appendix.

In order to compute likelihoods over data, a similar ODE is solved in the reverse direction. Starting from the target values $y_1^{\text{tgt}}$ at $t = 1$, samples are transformed back to $t = 0$ and their likelihood is evaluated with the standard Normal distribution. To accommodate for the change of volume, the change of variable formula is applied where the Jacobian is estimated across the probability path using the Hutchinson trace estimator.

**Running time** Our model is based on a transformer architecture, therefore it is interesting to analyze its running time compared to the TNP. First, the number of tokens used for predicting $N$ target points given a context of $M$ points in our model is $N + M$ compared to the TNP model which uses $2N + M$ in order to comply with causal masking. Second, generating samples with TNP requires $N$ evaluations as it is an autoregressive model. For our model the number of evaluations depends on the ODE solver and is a parameter that can be tuned according to the required accuracy, however it is independent of the number of target points $N$. The main disadvantage of our model compared to TNP is in evaluating likelihoods, where TNP requires only one model evaluation, and our model, similar to sampling, requires multiple evaluations as part of the ODE solution. Wall-clock time comparisons for some of the experiments are reported in Fig. 2 and Fig. 3.

## 3.3 Exchangeability and Consistency

In this section we discuss the properties of our model in relation to the requirements of the Kolmogorov extension theorem as presented in Sec. 2.1, namely the exchangeability and consistency properties.

Within a given conditional prediction task $p(y^{\text{tgt}}|x^{\text{tgt}}, \{x^{\text{ctx}}, y^{\text{ctx}}\})$, our model is guaranteed to be invariant with respect to permutations of both the context and the target, and therefore complies with the exchangeability property. This is due to the transformer architecture and the treatment of tokens as sets rather than sequences. Specifically, all tokens undergo the exact same processing and we do not use positional encodings that rely on ordering.

On the other hand, the consistency property is not implied by the architecture of our model and therefore it is not guaranteed. However, even though it is not guaranteed by the model itself, the training paradigm of NPs promotes both the marginal and conditional consistency properties. This is because in every training sample, the context and target sets are constructed randomly from a true underlying function, which is itself a sample from the stochastic process defined by the training set. Therefore the training objective is a Monte Carlo estimate of the ground truth stochastic process.

We note that also in previous NP models, consistency is not fully guaranteed and holds only in a limited sense. Specifically, while models like CNP, NP and TNP are consistent over marginalization (Eq. 4), they are not consistent over conditioning (Eq. 5). On the other hand models like NDP are consistent over conditioning and not over marginalization. For more discuison on this see App. A.

## 4 Experiments

Our experiments are conducted on three distinct data domains: synthetic 1D Gaussian processes, image data from EMNIST [10] and CelebA [28], and real-world weather prediction data ERA5 [16].

We implement a single FlowNP model architecture across all experiments in the main paper, adapting only the input dimensions $\dim(\mathcal{X})$ and output dimensions $\dim(\mathcal{Y})$. We use a transformer with 6 layers of full self attention, 128 hidden dimensions and 4 attention heads. We use sinusoidal encodings for the input $x$ and flow time $t$ with 10 frequencies per dimension, except for the ERA5 experiments where we use 40 frequencies per dimension. We emphasize that the encoding is a function of $x$ rather than the position in the sequence. For likelihood evaluation we use an ODE solver based on the midpoint method with 100 steps implemented by Lipman et al. [26], and for sampling we use the Euler method with 100 steps. See Alg. 1 and Alg. 2 in the appendix for the training and sampling algorithms respectively. The transformer architecture and implementation of our model are based on Nguyen & Grover [32]. All training, inference and sampling are performed with an NVIDIA RTX4090 GPU. The implementation of all experiments and models is available at https://github.com/danrsm/flowNP.

In the appendix we provide more examples of larger models trained on the CelebA image data and discuss a modified sampling method that incorporates small noise within the sampling process.

**Baselines** We compare to the baselines CNP [13], NP [14], ANP [21] and TNP [32] using the implementation in Nguyen & Grover [32] and Lee et al. [23]. In order to make a fair comparison of the main conceptual differences between the methods, we base our model on the transformer architecture in TNP, and reimplement NDP [11] using the same architecture. Therefore, both TNP and NDP models share our network architecture and differ only in the training objective and evaluation method. Specifically, TNP uses the same transformer as we do but is trained by autoregressive maximum likelihood using a causal mask as implemented by the authors, and for the NDP baseline we implement the model with the same transformer used in FlowNP and TNP, omitting the bi-directional attention used in the original NDP. Therefore, NDP differs from our model only by training on unconditional joint distributions, using a linear variance-preserving noise schedule and applying a diffusion loss based on predicting the denoised data rather than the flow velocity. In order to generate conditional samples from NDP we use a guidance-based method, and we evaluate the NDP likelihood by the probability flow ODE with the same ODE solver used for FlowNP. We note that for benchmarks that also appear on the original NDP paper (namely Fixed-Noisy RBF and Fixed-Noisy Matern) our implementation of NDP results in better log-likelihood than the original one.

### 4.1 Synthetic 1D functions

We start with the standard benchmarks of NPs generated from synthetic 1D Gaussian processes (GP). We follow the evaluation protocols of both Lee et al. [23] and Bruinsma et al. [8], which are used respectively in the TNP [32] and NDP [11] papers. In the first [23], we generate data from three different GP kernels: RBF (also called squared exponential - SE), Matern-$\frac{5}{2}$ and Periodic. For each kernel, functions are generated using parameters which are randomly chosen for each sampled function separately. Evaluation is done on held-out data using a random number of context and target points sampled uniformly between 3 and 47 where the total number is constrained to be equal or less than 50. In the second protocol [8] we use two GP kernels: RBF and Matern-$\frac{5}{2}$, using a *fixed* set of parameters, and additional Gaussian observation noise with variance $0.05^2$. Evaluation is done on

Table 1: Comparison of log-likelihood computed on the target set for various 1D GP datasets. We provide mean and standard deviation over five runs. FlowNP consistently achieves state-of-the-art performance, outperforming or matching all other models across all datasets.

| MODEL | RBF | MATERN-$\frac{5}{2}$ | PERIODIC | FIXED-NOISY RBF | FIXED-NOISY MATERN-$\frac{5}{2}$ |
|---|---|---|---|---|---|
| CNP | 0.31 ± 0.04 | 0.12 ± 0.02 | -0.63 ± 0.01 | -1.00 ± 0.02 | -1.09 ± 0.01 |
| NP | 0.31 ± 0.04 | 0.13 ± 0.05 | -0.61 ± 0.01 | -1.13 ± 0.02 | -1.18 ± 0.01 |
| ANP | 1.10 ± 0.04 | 0.85 ± 0.04 | -0.89 ± 0.04 | -0.87 ± 0.03 | -0.98 ± 0.02 |
| TNP | 1.65 ± 0.02 | 1.29 ± 0.02 | -0.58 ± 0.01 | 0.68 ± 0.01 | 0.30 ± 0.01 |
| NDP | 1.20 ± 0.04 | 0.94 ± 0.03 | -0.54 ± 0.01 | 0.71 ± 0.01 | 0.27 ± 0.02 |
| FLOWNP (OURS) | 1.69 ± 0.02 | 1.30 ± 0.02 | -0.50 ± 0.01 | 0.71 ± 0.01 | 0.30 ± 0.01 |

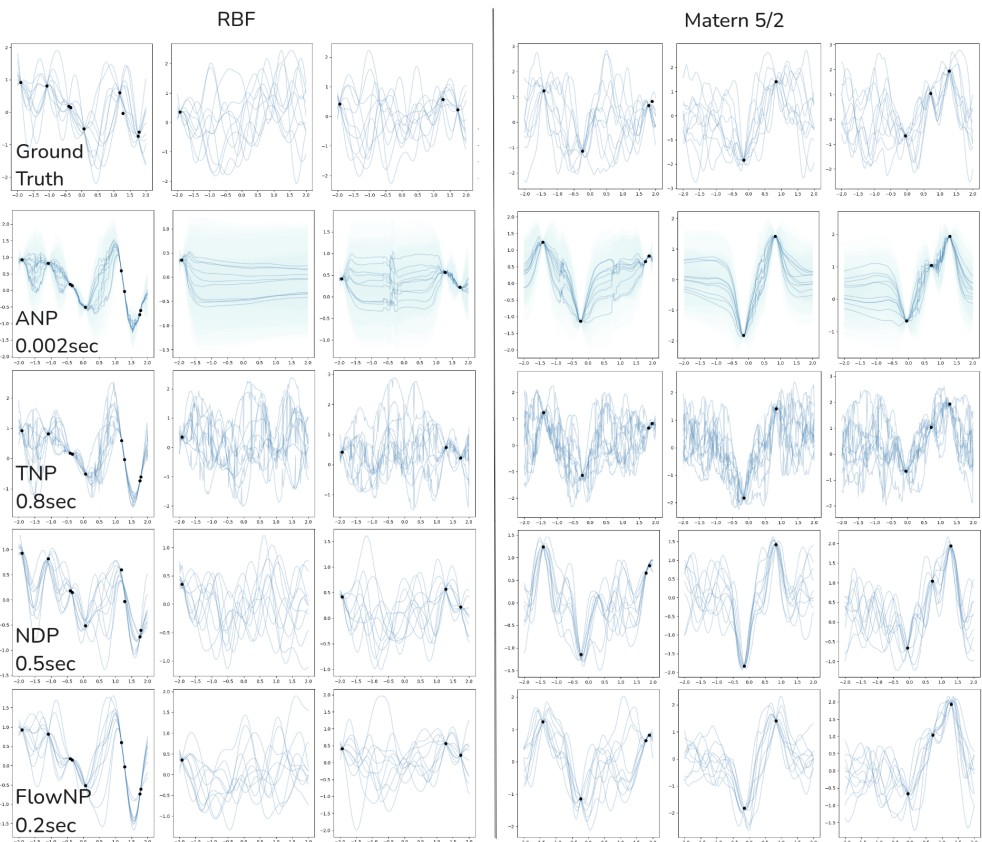

Figure 2: Samples from models trained on an RBF kernel (left) and a Matern-$\frac{5}{2}$ kernel (right). ANP captures the uncertainty mostly through its predicted local variance (shaded area) while TNP, NDP and FlowNP can generate coherent samples that cover the global uncertainty. In contrast to NDP, FlowNP generates conditional samples directly without needing auxiliary conditioning methods such as guidance. In contrast to TNP that generates the samples point-by-point, FlowNP generates all points in parallel, resulting in faster and smoother sampling.

held-out data with the same parameters, a uniformly random context size between 1 and 10, and a fixed number of 50 target points. These datasets are denoted by Fixed-Noisy RBF and Fixed-Noisy Matern-$\frac{5}{2}$. Likelihood for the latent variable models NP and ANP are computed with variational importance sampling [9] using 200 samples, and for the NDP and FlowNP using the midpoint ODE solver with 100 steps.

Results are summarized in Table 1, where for each dataset we report the mean and standard deviation of five repeated experiments, each using different random seeds and evaluation sets. Note that the performance of baseline models is improved compared to the previously reported results [32, 8] due to our hyperparameter tuning (see App. B). Our model, FlowNP, consistently outperforms all other models while in some cases achieving comparable results to the next best models, TNP and NDP.

Table 2: Target log-likelihood mean $\pm$ 1 standard deviation for the EMNIST and CelebA image datasets and the ERA5 weather dataset. FlowNP consistently outperforms all othe models.

| MODEL | EMNIST 0-9 | EMNIST 10-46 | CELEBA | ERA5 |
|---|---|---|---|---|
| CNP | $1.27 \pm 0.04$ | $0.73 \pm 0.04$ | $2.10 \pm 0.01$ | $4.06 \pm 0.06$ |
| NP | $1.17 \pm 0.03$ | $0.92 \pm 0.02$ | $2.53 \pm 0.03$ | $3.35 \pm 0.08$ |
| ANP | $1.35 \pm 0.02$ | $1.18 \pm 0.01$ | $3.24 \pm 0.01$ | $7.76 \pm 0.08$ |
| TNP | $2.08 \pm 0.02$ | $1.80 \pm 0.05$ | $3.95 \pm 0.02$ | $11.32 \pm 0.09$ |
| NDP | $1.58 \pm 0.02$ | $1.47 \pm 0.01$ | $4.28 \pm 0.04$ | $6.76 \pm 0.07$ |
| FLOWNP (OURS) | $2.50 \pm 0.01$ | $2.42 \pm 0.01$ | $6.37 \pm 0.01$ | $12.79 \pm 0.03$ |

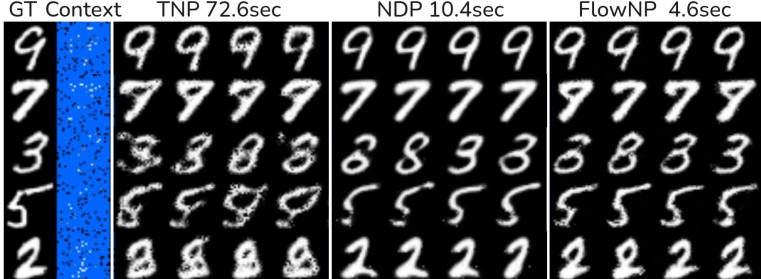

Figure 3: Conditional EMNIST samples generated by TNP, NDP and FlowNP that were trained only on subsets of pixels. For each model we show 4 samples. FlowNP generates sharp and diverse samples faster than other models.

Qualitative results are provided in Fig. 2, showing samples from models trained on the RBF and Matern kernels with fixed parameters. We compare ANP: a latent variable model trained with variational inference; TNP: an autoregressive model; NDP: a diffusion model of joint distributions; and our proposed model, FlowNP. ANP largely fails to capture the global uncertainty with the latent variable, and tends to attribute large variances to local uncertainty, depicted as the shaded area corresponding to the predicted variance of each point. In contrast, TNP, NDP and FlowNP generate plausible samples that cover the global uncertainty. However while TNP generates samples in an autoregressive manner sometimes resulting in a jumpy behavior, NDP and FlowNP generate all points in parallel mostly resulting in smoother samples. An advantage of FlowNP over NDP is that the conditional samples are generated directly from the model outpus rather than relying on a guidance method as used for NDP. For each method we note the wall-clock time for generating single samples, showing that FlowNP runs faster than TNP and NDP. For a visualiztion of the sampling process see Fig. 6 in the appendix.

## 4.2 Images

We follow with experiments on two image datasets, EMNIST [10] and CelebA [28], where the input $\mathcal{X}$ is 2-dimensional and represents the spatial position in the image, and the output $\mathcal{Y}$ is either a 1D grayscale value or a 3D RGB value of the pixels. The NP models are trained on random subsets of image pixels. For EMNIST, which contains images of characters, we train on the 10 first classes (digits 0-9) and evaluate both the in-distribution performance, and the out-of-distribution performance on the rest of the characters. Since the image data is generated from a discrete representation of the pixels, the likelihood of continuous models can grow arbitrarily (see e.g. App. B. in [24]). Therefore we add small noise with variance $0.01^2$ to the data.

Results are provided in Table 2. FlowNP outperforms all other models both for the in-distribution evaluation of EMNIST and CelebA, and for the out-of-distribution evaluation of EMNIST classes 10-46. We show samples generated by TNP, NDP and FlowNP of EMNIST images, using 70 observed pixels as context. Samples from FlowNP are sharp and diverse, and are generated by seven times less network evaluation and using less tokens compared to TNP. FlowNP samples also do not require computing guidance signals for conditioning like NDP. This leads to faster sampling.

In the appendix we show samples of CelebA images (Fig. 7) using a larger FlowNP model based on DiT-B [33] and compare to samples that are generated by adding small noise during the sampling process (App. C).

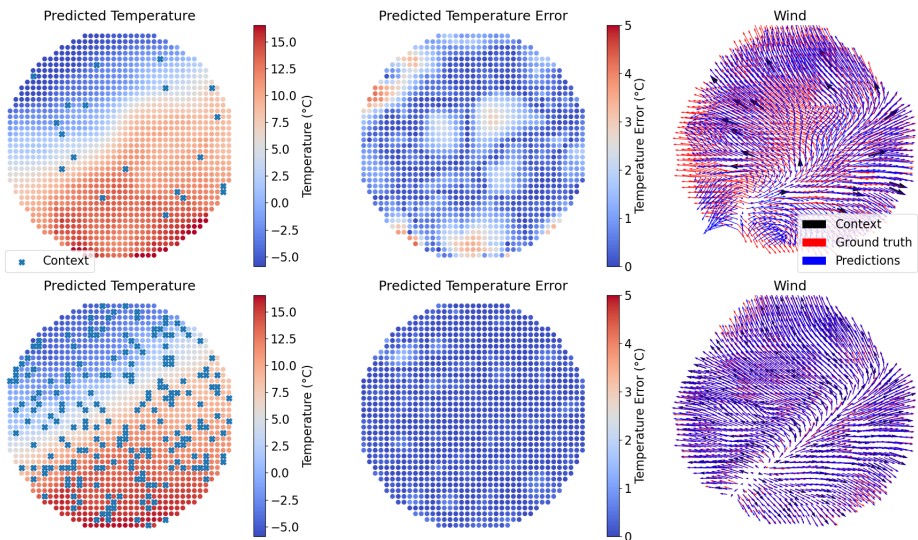

Figure 4: Visualization of temperature and wind prediction using conditional samples generated by FlowNP on two held-out data points from ERA5. The model generates coherent samples based on the context points. As more context points are given (bottom row) predictions become more accurate.

## 4.3 Weather data

To assess the performance of the FlowNP model in real-world dynamic and multi-dimensional tasks, we use meteorological data from the ERA5 global dataset [16]. We follow the setup in Holderrieth et al. [18] and extract data from a circular region with a 520km radius centered around Memphis, USA. Each sample represents a weather snapshot at a single point in time during the winter months in the years between 1980 to 2018 and consists of 1245 grid points of multiple meteorological variables: temperature, pressure, and two wind components (eastward and northward). We divide the years to 34K training samples and 17.5K evaluation samples. Training is performed using the NP setup where $\mathcal{X}$ represents the 2D longitude and latitude, $\mathcal{Y}$ represents the 4D meteorological variables and random subsets out of the 1245 points are used as the context and target sets.

The performance of FlowNP and the baselines are compared in Table 2, where the target set log-likelihood is reported for each model. FlowNP outperforms all other models.

For a qualitative evaluation, Figure 4 provides conditional samples of temperature and wind directions generated by FlowNP on the ERA5 weather data. The figure demonstrates how prediction quality scales with the size of the context set: the top row, using fewer context points, shows predictions with a greater degree of error compared to the bottom row, which is conditioned on a larger context set.

## 5 Analysis and Discussion

We provide analysis of FlowNP compared to the autoregressive approach of TNP and on different differentiating aspects compared to NDP. For more analysis on running times, different ODE solvers and the number of ODE steps for log-likelihood computation see App. D.

**FlowNP vs. TNP**   We consider the 1D step function, where the output changes from $y = 0$ to $y = 1$ at a random point of $x$. This function was used in Neal [31] and Dutordoir et al. [11] as an example that cannot be captured by Gaussian processes. Here we show that an autoregressive transformer-based model like TNP also cannot capture this function accurately. In Fig. 5 we show samples from TNP and FlowNP. While TNP samples are noisy in the transition area, FlowNP samples are sharp. The reason is that even though TNP can model complex distributions via the autoregression, it is constraind to Gaussians at every step. This is evident when looking at the marginal distribution at $x = 0$, shown on the right of Fig. 5 for both models. Comparing the marginal distribution of a single point with samples from the entire function indicates that, even without a formal guarantee, FlowNP's training approximately preserves the consistency property (Eq. 4).

Table 3: Target log-likelihood for ablations on three aspects differentiating FlowNP from NDP.

| Model | network output | noise schedule | conditioning | RBF | EMNIST |
|---|---|---|---|---|---|
| NDP | clean: $y_1$ | linear-vp: $\alpha_t = t, \beta_t = \sqrt{1-t^2}$ | unconditional | $1.20\pm0.04$ | $1.58\pm0.02$ |
| diffusion:clean | clean: $y_1$ | linear-vp: $\alpha_t = t, \beta_t = \sqrt{1-t^2}$ | conditional | $1.38\pm0.01$ | $1.64\pm0.01$ |
| diffusion:noise | noise: $y_0$ | linear-vp: $\alpha_t = t, \beta_t = \sqrt{1-t^2}$ | conditional | $1.33\pm0.04$ | $1.56\pm0.01$ |
| flow:lin-vp | velocity: $y_1 - y_0$ | linear-vp: $\alpha_t = t, \beta_t = \sqrt{1-t^2}$ | conditional | $0.41\pm0.02$ | $0.48\pm0.01$ |
| flow:poly2 | velocity: $y_1 - y_0$ | polynomial-2: $\alpha_t = t^2, \beta_t = (1-t)^2$ | conditional | $1.08\pm0.02$ | $1.60\pm0.02$ |
| flow:cosine | velocity: $y_1 - y_0$ | cosine: $\alpha_t = \sin(\frac{1}{2}\pi t), \beta_t = \cos(\frac{1}{2}\pi t)$ | conditional | $1.22\pm0.02$ | $1.80\pm0.01$ |
| flow:joint | velocity: $y_1 - y_0$ | cond-ot: $\alpha_t = t, \beta_t = 1 - t$ | unconditional | $1.73\pm0.04$ | $2.54\pm0.03$ |
| FlowNP | velocity: $y_1 - y_0$ | cond-ot: $\alpha_t = t, \beta_t = 1 - t$ | conditional | $1.69\pm0.02$ | $2.50\pm0.01$ |

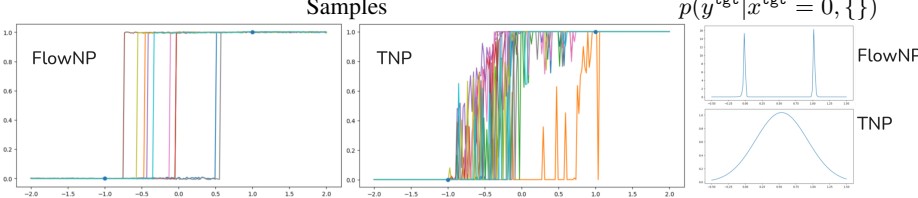

Figure 5: FlowNP vs. TNP for a random step function. FlowNP samples capture the sharp transition occurring in random positions, while TNP cannot model this function as each step in the autoregressive prediction is Gaussian. On the right: the marginal distribution of $y$ predicted for a single point where $x = 0$ further demonstrates this and highlights FlowNP's capacity to capture multimodal distributions.

**FlowNP vs. NDP**  We conduct an ablation study, summarized in Table 3, to investigate three key aspects differentiating FlowNP from the NDP baseline:

1. The output of the network at each step (clean target data $y_1$, noise $y_0$, or flow velocity $y_1 - y_0$)

2. The noise schedule used during training $y_t = \alpha_t y_1 + \beta_t y_0$

3. Amortizing context conditioning vs. modeling joint distributions only and computing conditional likelihoods through the joint/context ratio:

$$p\left(y^{\texttt{tgt}}|x^{\texttt{tgt}}, \{x^{\texttt{ctx}}, y^{\texttt{ctx}}\}\right) = p_\theta\left(y^{\texttt{tgt+ctx}}|x^{\texttt{tgt+ctx}}\right) / p_\theta\left(y^{\texttt{ctx}}|x^{\texttt{ctx}}\right)$$

Our results show that using the flow velocity as the network output and conditional optimal-transport noise schedule (cond-ot) yields superior performance across the models. We note that training FlowNP only on joint distributions (flow:joint) and using it to compute conditional likelihood through the joint/context ratio, achieves a slightly higher likelihood than a FlowNP model that is trained conditioned on contexts directly. This is perhaps expected as a model trained unconditionally dedicates its entire capacity to modeling the joint data distribution, a less demanding task than amortizing the conditioning on all possible context sets. However, this slightly higher likelihood comes at a cost of requiring auxiliary methods to generate conditional samples. In contrast, conditional training of FlowNP provides slightly lower maximum log-likelihood but is inherently capable of direct, auxiliary-free conditional sampling.

**Conclusion**  We presented FlowNP, a new model which implements neural processes using flow matching. This model is simple, efficient and outperforms previous neural processes models. We showed results on standard benchmarks such as 1D GP data and 2D images as well as real-world weather prediction data. Compared to NDP, we find that using a flow matching objective is favorable over diffusion and that modeling conditionals enables direct computation both for conditional likelihood and conditional sampling. Compared to TNP, FlowNP can efficiently capture non-Gaussian and multimodal distributions and generate samples for a set of points in parallel.

**Limitations**  The main limitation of FlowNP is the iterative sampling and likelihood computation. One possible approach to improve this could be based on methods like shortcut models [12] which explicitly train the model to require less ODE solver steps.

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

# A  Consistency of NP Models

We discuss the consistency property of different models in more detail. The original CNP and NP models use architectures that ensure that given a context, all target points are predicted independently. This implies that consistency over marginalization (Eq. 4) holds. However, the model structure does not guarantee consistency over conditioning (Eq. 5) because computing the model conditioned on different contexts cannot be guaranteed to lead to consistent results that comply with an underlying joint distribution.

TNP, which is based on a transformer performs best when it is used as an autoregressive model. In that case it does not comply neither with exchangeability nor consistency, since even when using clever masking to make the model invariant to the context order, the joint distribution of the target still depends on the ordering of the autoregression in the target points. While the authors of TNP propose different variants of the models that make it exchangeable and consistent with marginalization, they lead to a significant drop in performance, and are still not guaranteed to be consistent in terms of conditioning.

The reason that NDP is consistent over conditioning is that it only models joint distributions and therefore computes conditionals using the conditional definition:

$$p\left(y^{\texttt{tgt}}|x^{\texttt{tgt}}, \{x^{\texttt{ctx}}, y^{\texttt{ctx}}\}\right) = p_\theta\left(y^{\texttt{tgt+ctx}}|x^{\texttt{tgt+ctx}}\right)/p_\theta\left(y^{\texttt{ctx}}|x^{\texttt{ctx}}\right)$$

However the full self-attention architecture leads to predictions of target points that cannot be separated into independent factors and therefore cannot ensure consistency over marginalizations.

# B  Training and Sampling

The algorithms for training and sampling are provided in Alg. 1 and Alg. 2.

Fig. 6 provides a demonstration of the sampling process for 1D GP data by visualizing the transformation from random noise to samples from the conditional distribution of $p(y^{\texttt{tgt}}|x^{\texttt{tgt}}, \{x^{\texttt{ctx}}, y^{\texttt{ctx}}\})$.

The hyperparameters of the baselines are tuned to improve their performance. Namely, we find that for CNP and TNP no bound on the output variance is required, while for the models trained by variational inference, NP and ANP, we set the output variance bound to $0.05^2$ and use 8 importance samples for training. The implementation in `https://github.com/danrsm/flowNP` contains the configurations of all the models.

---

**Algorithm 1** Training

> **input:** dataset of functions $\mathcal{F}$
> **repeat**
>> $f_i \sim \mathcal{F}$ Sample a batch of functions, here shown for 1
>> $M, N \sim \mathcal{U}$ Random context and target sizes
>> $x^{\texttt{ctx}} \sim \mathcal{X}^M, x^{\texttt{tgt}} \sim \mathcal{X}^N$ Sample positions
>> $y^{\texttt{ctx}} \leftarrow f_i(x^{\texttt{ctx}}), y^{\texttt{tgt}} \leftarrow f_i(x^{\texttt{tgt}})$
>> $t \sim \mathcal{U}[0, 1]$ Sample flow time
>> $y_0^{\texttt{tgt}} \sim \mathcal{N}(0, I)$ Sample noise
>> $y_t^{\texttt{tgt}} \leftarrow t y^{\texttt{tgt}} + (1 - t) y_0^{\texttt{tgt}}$
>> $\texttt{tokens}^{\texttt{ctx}} = \{\texttt{embed}^\theta([x^{\texttt{ctx}}, 1, y^{\texttt{ctx}}])\}$ M tokens
>> $\texttt{tokens}^{\texttt{tgt}} = \{\texttt{embed}^\theta([x^{\texttt{tgt}}, t, y_t^{\texttt{tgt}}])\}$ N tokens
>> $\hat{u}_t \leftarrow u^\theta([\texttt{tokens}^{\texttt{ctx}}, \texttt{tokens}^{\texttt{tgt}}])$
>> $\mathcal{L}(\theta) = \|\hat{u}_t - (y^{\texttt{tgt}} - y_0^{\texttt{tgt}})\|^2$
>> $\theta \leftarrow \texttt{update}\left(\frac{\partial}{\partial \theta}\mathcal{L}(\theta)\right)$
> **until** convergence
> **output:** trained parameters $\theta$.

---

**Algorithm 2** Sampling

**input:** context $x^{\texttt{ctx}}, y^{\texttt{ctx}}$ and target positions $x^{\texttt{tgt}}$

$\texttt{tokens}^{\texttt{ctx}} = \{\texttt{embed}^\theta([x^{\texttt{ctx}}, 1, y^{\texttt{ctx}}])\}$ M tokens

$y^{\texttt{tgt}} \sim \mathcal{N}(0, I)$ Random initialization

**for** $t = 0$ **to** $1$ in $\delta = 1/\texttt{n\_steps}$ increments **do**

    $\texttt{tokens}^{\texttt{tgt}} = \{\texttt{embed}^\theta([x^{\texttt{tgt}}, t, y^{\texttt{tgt}}])\}$ N tokens

    $\hat{u}_t \leftarrow u^\theta([\texttt{tokens}^{\texttt{ctx}}, \texttt{tokens}^{\texttt{tgt}}])$

    $y^{\texttt{tgt}} \leftarrow y^{\texttt{tgt}} + \delta \hat{u}_t$

**end for**

**output:** predicted sample $y^{\texttt{tgt}}$.

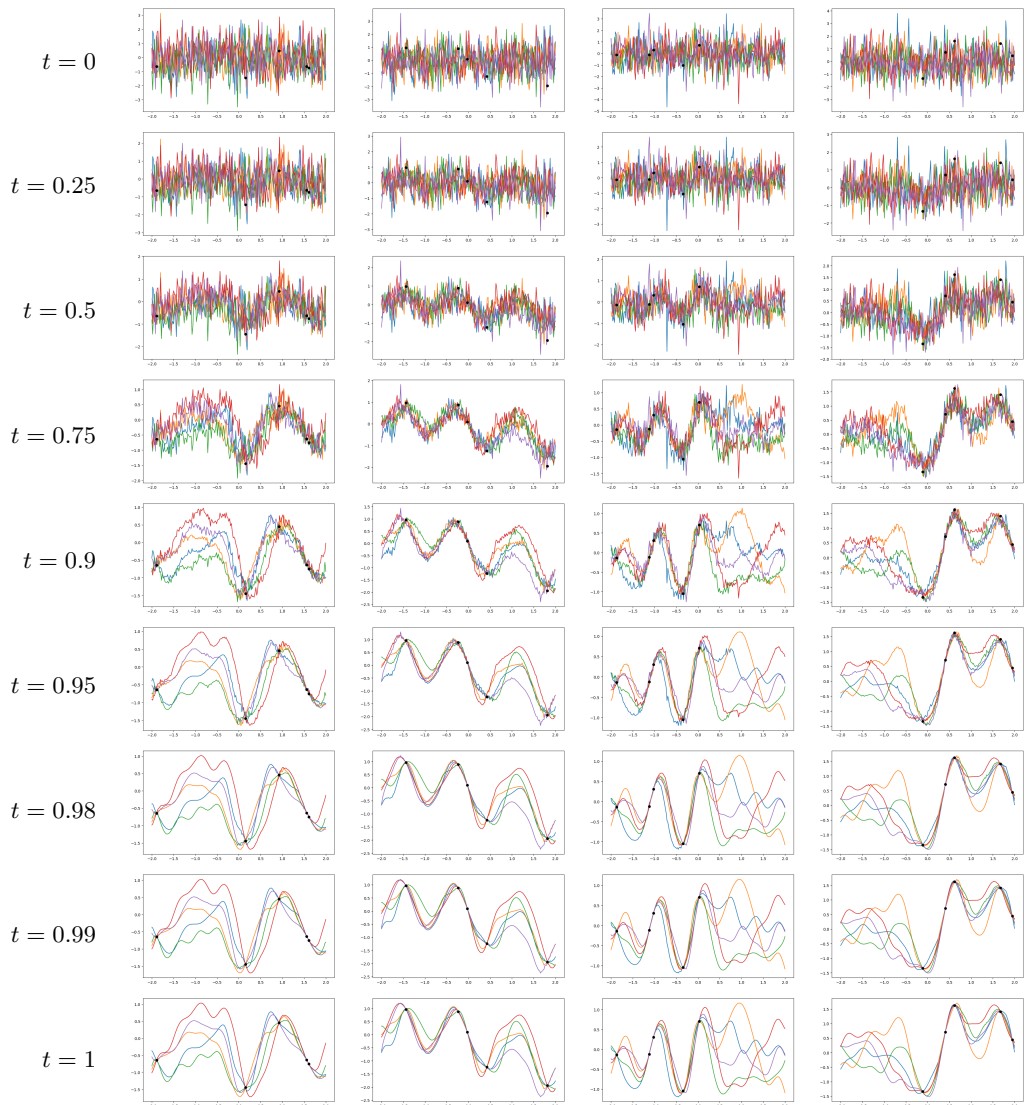

Figure 6: A visualization of the sampling process, showing the intermediate values $y_t^{\texttt{tgt}}$ for different time steps. Each column is conditioned on a different context and each color represents a different random sample generated by FlowNP.

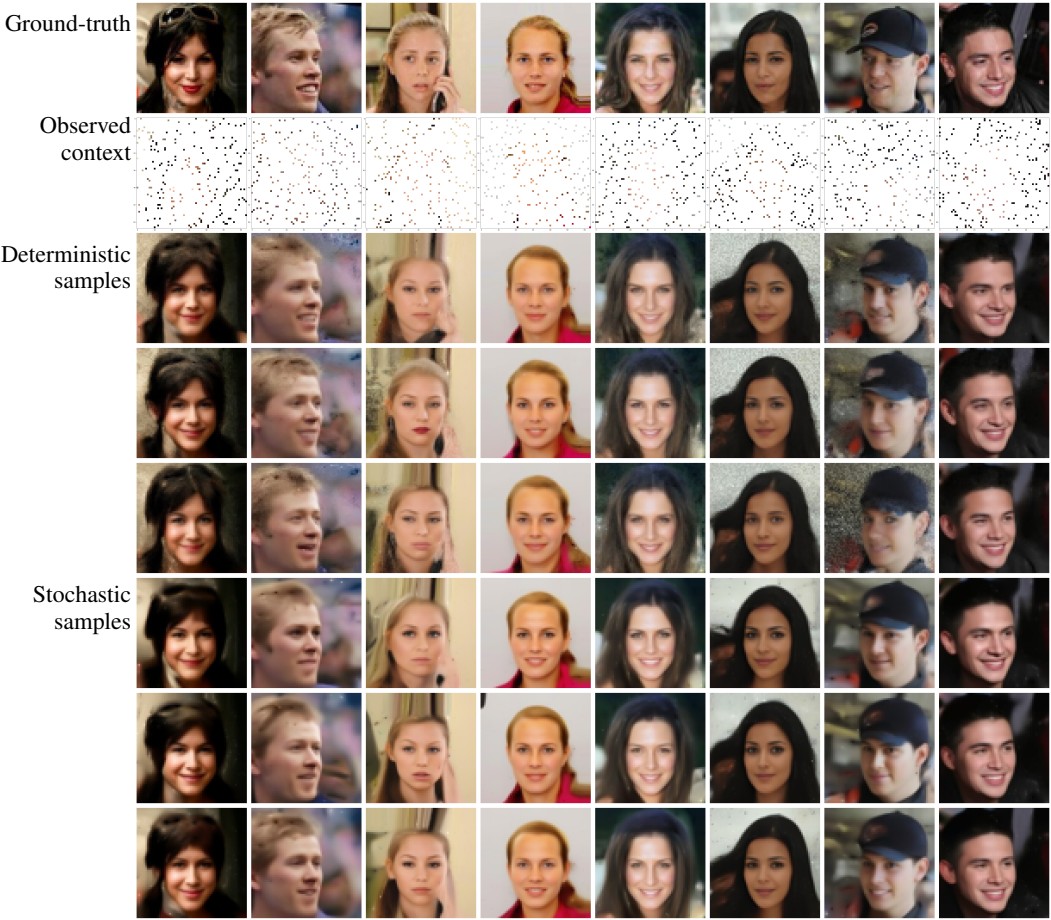

Figure 7: Generating conditional samples of CelebA data using the FlowNP model trained only on subsets of pixels.

## C    CelebA Images

We train a larger FlowNP model using the DiT-B architecture [33] on CelebA images with a resolution of $64 \times 64$. The model consists of 12 transformer layers with 768 hidden dimensions and 12 attention heads. We train this model using the same approach as in the main paper, by sampling random subsets of context pixels and target pixels every time an image is drawn. We the use this model to generate samples conditioned on a context of 5% observed pixels as shown in Fig. 7.

In some cases, we find that adding small levels of noise and scaling the predicted velocity during the sampling process results in samples that look more coherent and less noisy. We implement this by replacing the update of $y^{\texttt{tgt}}$ in each step of Alg. 2 to the following:

$$y^{\texttt{tgt}} \leftarrow y^{\texttt{tgt}} + \delta \alpha_t \hat{u}_t + \delta \sigma_t \nu \tag{11}$$

Where $\nu \sim \mathcal{N}(0, I)$ is random noise, $\alpha_t$ are values greater than 1 and $\sigma_t$ are small values. In our implementation we use $\alpha_t = 1 + t(1 - t)$ and $\sigma_t = 0.2t^2(1 - t)^2$.

A comparison between generated samples using Alg. 2 (Deterministic samples) and samples generated with the noise modification (Stochastic samples) is provided in Fig. 7. While the results are very similar, it can be seen that the deterministic samples sometimes appear noisier.

This approach did not make any difference for the GP data which suggests that it is effective when the capacity of the model is limited relative to the complexity of the distribution.

| ODE-solver | log-likelihood |
| --- | --- |
| Euler | 1.7941 |
| Midpoint | 1.6890 |
| RK4 | 1.6856 |
| Dopri5 | 1.6857 |

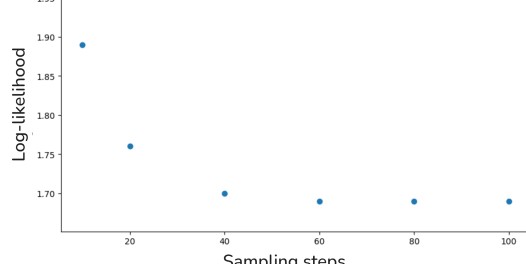

Table 4: Comparing different ODE solvers on the RBF GP data.

Figure 8: Log-likelihood as a function of number of ODE steps, computed using the midpoint ODE solver on the RBF GP data.

## D   Analysis

**ODE solver**   We compute the likelihood using several different ODE solvers for the FlowNP model on the RBF experiment. The results, presented in Table 4, show that except for the Euler method which overestimates the likelihood, all other solvers compute very similar values.

**Number of ODE steps**   We run an ablation on sampling and log-likelihood evaluation for different numbers of steps in the ODE solver using the midpoint method. For GP-RBF data, the mean of 5 random seeds with different number of steps is presented in Fig. 8. The std of for all is 0.014. This shows that the evaluation plateaus after T=60 steps. We observe similar behavior for sample quality.

**Running time**   Comparing the sampling time between FlowNP, NDP and TNP using an NVIDIA RTX4090 GPU and 100 sampling steps for FlowNP: the time to generate 1 sample in the GP experiment with 200 target points is 0.2sec for FlowNP, 0.5sec for NDP and 0.8sec for TNP. The time to generate 1 EMNIST sample with 748 target points is 4.6sec for FlowNP, 10.4sec for NDP and 72.6sec for TNP.

