# OpenReview forum: "Flow Matching Neural Processes"
_NeurIPS.cc/2025/Conference — NeurIPS 2025 poster_

### Official Review · Reviewer_KkGa · 2025-06-01

**Clarity:** 3
**Significance:** 2
**Originality:** 2
**Rating:** 4
**Confidence:** 4

**Summary:**

This paper proposes a flow matching neural process (FlowNP), which combines the neural process with flow matching by using flow matching to generate target value given target position and context. The proposed FlowNP is simple to implement and can sample from conditional distributions without auxiliary methods. Moreover, the model provides a controllable tradeoff between accuracy and running time via ODE solver steps. Empirically, they show that FlowNP outperforms SOTA in 1D GP, 2D image and weather data.

**Questions:**

See my question in weakness: what’s the gain for overlaying FM with NP? Can it significantly beat FM, or give us some insight/ inspiration for new methods?

**Ethical Concerns:**

["NO or VERY MINOR ethics concerns only"]

**Final Justification:**

The author adjusted main issues I proposed.

**Limitations:**

The paper already discussed their limitations: iterative sampling and likelihood computation.

**Paper Formatting Concerns:**

No formatting concerns

**Quality:**

3

**Strengths And Weaknesses:**

**Strength**: the idea is simple and show improvement from previous NP methods. The writing is clear and very easy to follow.

**Weakness**: The proposed method combines FM and NP in a straightforward way, which may raise questions regarding novelty. However, if it works fine, I’m happy to solve problems in a simple way. But I’m not sure if overlaying FM with NP is necessary: what’s the benefit of FlowNP, comparing to using FM directly (FM is flexible enough, and can be used for inference, sampling and conditional sampling)? If the improvement is not very significant but can potentially give insights/ inspirations to new methods, I’m also OK with it.

---

> ### Author Rebuttal · Authors · 2025-07-30
>
> We thank the reviewers for their thorough review and constructive feedback, and we are happy that most reviewers recommend accepting our work.
>
> We address all the issues raised by the reviewers and as requested, include more detailed explanations and additional results that answer reviewers’ questions and further support our claims (see summary at the end of this response). We hope that the reviewers will acknowledge this.
>
> ## Flow matching + NP vs. Flow matching
> The main motivation for using flow matching within the neural processes (NP) framework as we propose is to have a flexible model of signals over continuous spaces. In contrast standard diffusion or flow matching models that are trained on data with specific structure and dimension (e.g. a grid of pixels for images), NP models operate on continuous signals and can be used to generate samples or compute probabilistic inference involving arbitrary points within the signal's space.  In that regard, the experiments over images, although a standard NP benchmark, are a bit misleading since they demonstrate the performance on functions that only have values in regular gird intervals. Note that in contrast to standard diffusion or flow matching models of images, NP models only have access to a random number of pixels per image during training and never see a whole image. We will make this point clearer in the text (see also response to reviewer ZPkJ weakness 1).
>
>  We believe that within the framework of NPs, our work of introducing flow matching provides significant contributions compared to previous models including the implementation and conceptual simplicity and the ability to capture more complex distributions without relying on autoregressive modeling,
>
> # Summary of additional results (repeated in all responses):
> As requested, we have produced additional results to answer the reviewers’ questions and further support our claims. Some results include images that demonstrate qualitative aspects of samples, and will be added to the revised paper.
>
>
> We summarize the additional results:
>
> ### 1. Training with different flow schedules and computing likelihood with different ODE solvers.
>
> We have trained our model using various schedules for the RBF GP data. Each model is then evaluated on held out data by computing its log-likelihood. Our results are as follows (using the general schedule of $x_t = \alpha_t x_1 +  \beta_t x_0$ with different choices of $\alpha_t$ and $\beta_t$):
>
> | Name        | Schedule                                     | Log-Likelihood |
> |:------------|:---------------------------------------------|---------------:|
> | CondOT      | $\alpha_t = t, ~~ \beta_t = 1- t$            |         1.6890 |
> | Polynomial2 | $\alpha_t = t^2, ~~ \beta_t = (1- t)^2$     |         1.0955 |
> | Cosine      | $\alpha_t  = \sin(0.5 t \pi), ~~ \beta_t = \cos(0.5 t \pi)$            |         1.2441 |
> | LinearVP    | $\alpha_t = t, ~~ \beta_t = \sqrt{1- t^2}$            |         0.4112 |
>
>
> The results show schedules other than the standard CondOT perform worse in this setting.
>
> We also experiment with different ODE solvers for computing the log likelihood. For this we use our model trained with CondOT on RBF. We use a few different solvers using the implementation in the flow_matching package (based on torchdiffeq). The resulting  log likelihoods are:
>
> | ODE Solver | Log-Likelihood |
> |:-----------|---------------:|
> | Euler      |         1.7941 |
> | Midpoint   |         1.6890 |
> | RK4        |         1.6856 |
> | Dopri5     |         1.6857 |
>
> The results show that except for the simple Euler method, other methods result in very similar values.
> We will add both results above together with the same results on EMNIST to table 4 in the appendix to complete the ablation results.
>
> ### 2. Higher resolution CelebA samples
>
> We have trained our model on higher resolution CelebA images $64 \times 64$ and generated high quality conditional and unconditional samples. We will add these results to figure 6 in the appendix  (in the paper we showed results only for $32 \times 32$ resolution, following the experimental protocol of prior models).
>
>
> ### 3. Deterministic vs. stochastic sample generation
>
> We have added a qualitative comparison of deterministic sample generation (using an ODE solver), vs. a method that adds small amount of noise in the sampling process. As we mention in the paper, for the GP and ERA5 datasets we did not see any difference between the methods, however for image datasets (especially CelebA) the samples are a bit noisier in the deterministic case. This effect is even less visible in the larger model trained for resolution $64 \times 64$ (see additional results #2) where the deterministic and stochastic samples are very similar.
>
> ### 4. Error bars for ERA5
>
> We have repeated the model training for ERA5 data with 5 different seeds, and obtained a mean and std of 11.6647 +- 0.2452.

---

> > ### Comment · Reviewer_KkGa · 2025-08-08
> >
> > Thanks the author for providing detailed answers. Most my concerns are addressed, and I had raised my rating.

---

### Official Review · Reviewer_ZPkJ · 2025-06-11

**Clarity:** 3
**Significance:** 2
**Originality:** 2
**Rating:** 4
**Confidence:** 3

**Summary:**

The authors propose to train a model in the neural process framework with the flow matching algorithm. In their experiments, they outperform other neural process models on in-painting tasks for images and weather data and in terms of test-set likelihood.

**Questions:**

1. Line 190: Why does TNP require $2N + M$?
2. Figure 2: Which kernels are in which column? What do true samples look like for reference?
3. Do I understand Appendix A correctly that FlowNP is conditionally consistent but not marginally consistent?
4. What is $\alpha_t$ in Algorithm 2?
5. How well does your model perform on unconditional image generation? Would it reduce to flow matching?

**Ethical Concerns:**

["NO or VERY MINOR ethics concerns only"]

**Final Justification:**

Why not higher: While the method is interesting, I don't classify it as "high impact" at this point.

Why not lower: The paper reads well and has no major flaws.

**Limitations:**

Yes

**Quality:**

2

**Strengths And Weaknesses:**

The paper is easy to follow and the authors evaluate their proposed model on a variety of data types.

Weaknesses:
1. The abstract claims to outperform state-of-the-art methods on various benchmarks but only includes neural process models.
2. Line 192: The claim that "it is independent of the number of target points $N$" is slightly misleading. While the number of function evaluations is independent of it, the run time of each individual function evaluation does scale with $N$.
3. Line 273: The authors claim that FlowNP is the fasted method in Figure 2. However, ANP is actually the fastest by a factor of 100.
4. Table 2 and Line 281: You report continuous log-likelihoods, which is why they can grow without bounds. However, the standard for images would be to report negative log-likelihoods of the discretized data in bits per dimension, see for example Appendix B in [1] for an explanation. Then, your numbers could also be compared to other models directly, e.g. diffusion models or neural ODEs.
5. Figure 4 does not include true data for reference.
6. Both data and samples of CelebA in Figure 7 are much blurrier than they should be. Did you downsample the data? If so, did you make sure to plot the images at the same resolution that they actually are? If I zoom in, it looks like a blurry high-resolution image.

Remarks:
1. In Section 2.1, all $y_{i:n}$ should be $y_{1:n}$, I believe.
2. The caption of Figure 2 has a typo in "condinting".

[1] Lienen, Kollovieh and Günnemann. "Generative Modeling with Bayesian Sample Inference.", https://arxiv.org/abs/2502.07580

---

> ### Author Rebuttal · Authors · 2025-07-30
>
> We thank the reviewers for their thorough review and constructive feedback, and we are happy that most reviewers recommend accepting our work.
>
> We address all the issues raised by the reviewers and as requested, include more detailed explanations and additional results that answer reviewers’ questions and further support our claims (see summary at the end of this response). We hope that the reviewers will acknowledge this.
>
> ## Comparison to non-NP models (W1)
> **W1:** We only claim our method is advantageous against previous Neural Processes (NP) models. We agree that this can be presented more clearly and will update the abstract and introduction accordingly. As NPs are models of signals over continuous spaces, it is not directly clear how to compare them to models of gridded data such as standard diffusion models for images. We experiment with images (as is standard in NP papers) because they present an interesting example of complex functions over 2D spaces. The fact that the values of these functions (images) are only available in a regular grid is actually misleading from the main motivation of NP models (which is modeling continuous space signals). Note that during training, NP models only access a random number of pixels and never see a full image. We will emphasize this point further in the paper.
>
> ## Running time analysis (W2, W3, Q1)
> **W2:** We agree that the number of target points affect the running time of each feed forward through the network (network evaluation). We separated our analysis to (1) the running time of each network evaluation (by comparing the number of tokens) and (2) the required number of network evaluations. When comparing to the TNP baseline which is implemented with the same transformer architecture as ours, we explicitly state the number of tokens used in each feed-forward (lines 189-190). Our model uses N+M tokens (number of target points + number of context points) vs. 2*N+M for the TNP. **Q1:** The reason TNP has an additional token per target is because of the method they apply to incorporate causal masks among the target points - each target point is presented once with the coordinate only, $x$, and once as a pair of coordinates and values $x, y$.
>
> **W3:** In figure 2 our method is the fastest among the top performing methods, TNP and NDP, which form the main baselines we compare to. We apologize for the mistake (we only included ANP at a later stage) and have fixed line 273 to state “FlowNP is faster than TNP and NDP”.
>
> ## Continuous log likelihoods (W4)
> **W4:** The reason we do not measure the log-likelihood in bits per pixels is that NP models (ours and the baselines) are models of signals over continuous spaces (see also answer to W1). This means we effectively model infinite pixels, even when we measure the performance on regular-gridded images.
> When the values in the data are obtained from an original discrete representation (like the EMNIST and CELEBA images) we add small amount of noise to prevent arbitrary growing log-likelihood. We will further emphasize this point and include a discussion that compares this to the setup described in [1].
>
>
> [1] Lienen, Kollovieh and Günnemann. "Generative Modeling with Bayesian Sample Inference.", https://arxiv.org/abs/2502.07580
>
>
> ## CelebA qualitative results (W6, Q5)
> **W6:** All the CelebA experiments including the images presented in the appendix are for a downsampled $32 \times 32$ resolution, because we follow the experimental setup of the previous NP methods used as baselines. We have now trained a FlowNP model on higher resolution $64 \times 64$. The generated samples are of high quality and we will include them in the appendix of the revised paper.
>
> **Q5:** Using this model, we have also generated unconditional images (by giving zero context points). The quality and diversity of the samples demonstrate the capacity of the model to capture both conditional and unconditional distributions of complex signals. This model still does not reduce to standard flow matching on images as it models signals on continuous spaces as demonstrated on the 1D GP data in the paper and discussed in W1 in this response.
>
>
> ## Other points:
>
> **W5:** The blue arrows are the ground truth values at each point and are to be compared with the red arrows which are the model’s predictions.
>
> **Q2:** In figure 2 the RBF kernel is on the left and Matern Kernel is on the right. We will add this to the caption and include the GT samples using Gaussian Processes. Thank you for pointing this out.
>
> **Q3:** In appendix A we discuss previous NP models. In section 3.3 we discuss the conditions of Kolmogorov Extension Theorem (defined in Sec. 2.1) for our model. In short - exchangeability is guaranteed and both conditional consistency and marginal consistency are not guarnateed.
>
> **Q4:** $\alpha_t$ is a scalar close to 1 that adapts the step size in the samplng process. See lines 438-440, and discussion with reviewer qfUz about sampling.
>
> **Remarks 1 & 2:**  We will fix these typos. Thank you for pointing them out.
>
>
> # Summary of additional results (repeated in all responses):
> As requested, we have produced additional results to answer the reviewers’ questions and further support our claims. Some results include images that demonstrate qualitative aspects of samples, and will be added to the revised paper.
>
>
> We summarize the additional results:
>
> ### 1. Training with different flow schedules and computing likelihood with different ODE solvers.
>
> We have trained our model using various schedules for the RBF GP data. Each model is then evaluated on held out data by computing its log-likelihood. Our results are as follows (using the general schedule of $x_t = \alpha_t x_1 +  \beta_t x_0$ with different choices of $\alpha_t$ and $\beta_t$):
>
> | Name        | Schedule                                     | Log-Likelihood |
> |:------------|:---------------------------------------------|---------------:|
> | CondOT      | $\alpha_t = t, ~~ \beta_t = 1- t$            |         1.6890 |
> | Polynomial2 | $\alpha_t = t^2, ~~ \beta_t = (1- t)^2$     |         1.0955 |
> | Cosine      | $\alpha_t  = \sin(0.5 t \pi), ~~ \beta_t = \cos(0.5 t \pi)$            |         1.2441 |
> | LinearVP    | $\alpha_t = t, ~~ \beta_t = \sqrt{1- t^2}$            |         0.4112 |
>
>
> The results show schedules other than the standard CondOT perform worse in this setting.
>
> We also experiment with different ODE solvers for computing the log likelihood. For this we use our model trained with CondOT on RBF. We use a few different solvers using the implementation in the flow_matching package (based on torchdiffeq). The resulting  log likelihoods are:
>
> | ODE Solver | Log-Likelihood |
> |:-----------|---------------:|
> | Euler      |         1.7941 |
> | Midpoint   |         1.6890 |
> | RK4        |         1.6856 |
> | Dopri5     |         1.6857 |
>
> The results show that except for the simple Euler method, other methods result in very similar values.
> We will add both results above together with the same results on EMNIST to table 4 in the appendix to complete the ablation results.
>
> ### 2. Higher resolution CelebA samples
>
> We have trained our model on higher resolution CelebA images $64 \times 64$ and generated high quality conditional and unconditional samples. We will add these results to figure 6 in the appendix (in the paper we showed results only for  $32 \times 32$ resolution, following the experimental protocol of prior models).
>
>
> ### 3. Deterministic vs. stochastic sample generation
>
> We have added a qualitative comparison of deterministic sample generation (using an ODE solver), vs. a method that adds small amount of noise in the sampling process. As we mention in the paper, for the GP and ERA5 datasets we did not see any difference between the methods, however for image datasets (especially CelebA) the samples are a bit noisier in the deterministic case. This effect is even less visible in the larger model trained for resolution $64 \times 64$ (see additional results #2) where the deterministic and stochastic samples are very similar.
>
> ### 4. Error bars for ERA5
>
> We have repeated the model training for ERA5 data with 5 different seeds, and obtained a mean and std of 11.6647 +- 0.2452.

---

> > ### Comment · Reviewer_ZPkJ · 2025-08-01
> >
> > Thank you for clarifying the claims and the interesting extra information. With misleading claims out of the way, I will recommend acceptance.
> >
> > In case you are using matplotlib, remember to plot images with `interpolation="none"` to render them at their true resolution.

---

### Official Review · Reviewer_KAXg · 2025-06-26

**Clarity:** 2
**Significance:** 2
**Originality:** 2
**Rating:** 4
**Confidence:** 2

**Summary:**

The paper introduces FlowNP, a novel Neural Process (NP) model that uses flow matching for learning stochastic processes from data. FlowNP is designed to address limitations of existing NP models, such as underfitting, sequential sampling inefficiencies in autoregressive models, and lack of global uncertainty representation. The model uses a transformer architecture to predict velocity vectors for target points, allowing parallel sampling via an ODE solver. Experiments are provided to demonstrate that FlowNP outperforms existing NP models in terms of log-likelihood and sampling efficiency.

**Questions:**

- Can FlowNP be used to provide quantification of epistemic uncertainty?
- Is FlowNP robust to noise perturbation in the training datasets?
- Does FlowNP suffer from spectral bias (over-smoothing high frequencies)?
- What are the limitations of the proposed method? For what data regimes the method excel and fail?

**Ethical Concerns:**

["NO or VERY MINOR ethics concerns only"]

**Final Justification:**

My final decision is a borderline accept. Overall the paper is a neat contribution to NeurIPS, providing a principled method to integrate flow matching with NPs. The authors have also sufficiently addressed my concerns and included additional results to help strengthen the paper. Provided that these additional results are included, I am inclined to go with a borderline accept.

**Limitations:**

Yes

**Quality:**

2

**Strengths And Weaknesses:**

Strengths:
- The integration of flow matching with NPs is interesting, providing a new way to model conditional distributions without relying on autoregressive or auxiliary conditioning methods
- The proposed parallel sampling method improves efficiency during generation, as it avoids sequential generation of autoregressive approaches

Weaknesses:
- Lack of consistency guarantees, since consistency (marginal and conditional) is not formally guaranteed
- Lack of scalability evaluation, as it is not clear how well the method would scale to settings with very large context/target sets (e.g., data with higher resolutions)
- Lack of ablation study on the design of the flow matching framework (sensitivity to choice of noise schedule/probability path, choice of ODE sampler, etc.)

---

> ### Author Rebuttal · Authors · 2025-07-30
>
> We thank the reviewers for their thorough review and constructive feedback, and we are happy that most reviewers recommend accepting our work.
>
> We address all the issues raised by the reviewers and as requested, include more detailed explanations and additional results that answer reviewers’ questions and further support our claims (see summary at the end of this response). We hope that the reviewers will acknowledge this.
>
> ## Choice of scheduling and solver
> We thank the reviewer for mentioning this. The simplicity of switching between different noise schedules and ODE solver is one of the benefits of the flow matching framework. We agree that demonstrating results with more choices will improve the paper. We therefore added more experiments by training models with various noise schedules, and evaluating the model likelihood with different ODE solvers. See additional results #1. The results will be added to table 4 in the appendix to complete the ablation analysis. In general the results show that the standard conditional optimal transport schedule works best in our 1D and 2D settings (although for more specialized settings other schedules might work better). Regarding the ODE solvers for likelihood computation, we find that all tested solvers behave very similarly, except for the Euler solver.
> For sampling, we qualitatively compared samples generated with the Euler and midpoint solvers and could not find visible differences between them.
>
> ## Lack of consistency guarantees
>
> Consistency guarantees is one of the big open challenges in NP models. While some simple models can achieve some level of guarantees, developing a high capacity model (e.g. without independent assumptions like CNP) is an active area of research. We hope our paper can serve as a good starting point for this.
>
> ## Lack of scalability evaluation
>
> While we provide running time analysis that shows FlowNP is favorable compared to the baselines, we agree that our experiments are relatively small scale. Scaling up NPs is an important next frontier in NP research, and the results we add for higher resolution of CelebA as requested by reviewer ZPkJ (see additional results #2) is perhaps a first small step in this direction.  We believe that weather data provides an excellent opportunity to develop and evaluate large scale modeling.
>
> ## Answer to questions
>
> **Epistemic uncertainty in NP:**
> We are not aware of methods to compute epistemic uncertainty in NP models. This could be an interesting future research direction.
>
> **Robustness to noise:**
> While some of our datasets add observational noise we do not explicitly test the effect it has. This could also be an interesting future research direction.
>
> **Spectral bias:**
> We do not test this quantitatively but we do observe that samples (specially in images) are generally smoother than the GT signal. We believe this is usually the case for models trained with mean-seeking objectives such as maximum likelihood and flow matching.
>
> **Regimes where FlowNP excels / fails:**
> Compared to TNP, FlowNP would work better when the data is highly non Gaussian and there is no natural order of points. Otherwise, autoregressive models that use Gaussian distributions would be better.  Compared to NDP - while FlowNP  outperforms NDP in the settings that we test, we are not sure how both models would behave in different settings. Related also to the above discussion about different noise schedules, it would be interesting to find specific aspects in the signals that make different schedules work better than others.
>
>
> # Summary of additional results (repeated in all responses):
> As requested, we have produced additional results to answer the reviewers’ questions and further support our claims. Some results include images that demonstrate qualitative aspects of samples, and will be added to the revised paper.
>
>
> We summarize the additional results:
>
> ### 1. Training with different flow schedules and computing likelihood with different ODE solvers.
>
> We have trained our model using various schedules for the RBF GP data. Each model is then evaluated on held out data by computing its log-likelihood. Our results are as follows (using the general schedule of $x_t = \alpha_t x_1 +  \beta_t x_0$ with different choices of $\alpha_t$ and $\beta_t$):
>
> | Name        | Schedule                                     | Log-Likelihood |
> |:------------|:---------------------------------------------|---------------:|
> | CondOT      | $\alpha_t = t, ~~ \beta_t = 1- t$            |         1.6890 |
> | Polynomial2 | $\alpha_t = t^2, ~~ \beta_t = (1- t)^2$     |         1.0955 |
> | Cosine      | $\alpha_t  = \sin(0.5 t \pi), ~~ \beta_t = \cos(0.5 t \pi)$            |         1.2441 |
> | LinearVP    | $\alpha_t = t, ~~ \beta_t = \sqrt{1- t^2}$            |         0.4112 |
>
>
> The results show schedules other than the standard CondOT perform worse in this setting.
>
> We also experiment with different ODE solvers for computing the log likelihood. For this we use our model trained with CondOT on RBF. We use a few different solvers using the implementation in the flow_matching package (based on torchdiffeq). The resulting  log likelihoods are:
>
> | ODE Solver | Log-Likelihood |
> |:-----------|---------------:|
> | Euler      |         1.7941 |
> | Midpoint   |         1.6890 |
> | RK4        |         1.6856 |
> | Dopri5     |         1.6857 |
>
> The results show that except for the simple Euler method, other methods result in very similar values.
> We will add both results above together with the same results on EMNIST to table 4 in the appendix to complete the ablation results.
>
> ### 2. Higher resolution CelebA samples
>
> We have trained our model on higher resolution CelebA images $64 \times 64$ and generated high quality conditional and unconditional samples. We will add these results to figure 6 in the appendix  (in the paper we showed results only for $32 \times 32$ resolution, following the experimental protocol of prior models).
>
>
> ### 3. Deterministic vs. stochastic sample generation
>
> We have added a qualitative comparison of deterministic sample generation (using an ODE solver), vs. a method that adds small amount of noise in the sampling process. As we mention in the paper, for the GP and ERA5 datasets we did not see any difference between the methods, however for image datasets (especially CelebA) the samples are a bit noisier in the deterministic case. This effect is even less visible in the larger model trained for resolution $64 \times 64$ (see additional results #2) where the deterministic and stochastic samples are very similar.
>
> ### 4. Error bars for ERA5
>
> We have repeated the model training for ERA5 data with 5 different seeds, and obtained a mean and std of 11.6647 +- 0.2452.

---

> > ### Comment · Reviewer_KAXg · 2025-08-02
> > **Thank you for the rebuttal**
> >
> > I thank the authors for the rebuttal. The responses have mostly addressed my concerns. I am inclined to accept and keep my score.

---

### Official Review · Reviewer_qfUz · 2025-06-30

**Clarity:** 3
**Significance:** 3
**Originality:** 2
**Rating:** 5
**Confidence:** 3

**Summary:**

- The authors use Flow Matching to improve sample generation from Neural Processes.
- The authors use the known ability of transformers to aggregate context data points very effectively.
- The authors use Flow matching as a method to generate coherent samples with more expressive densities than Transformer Neural Processes and better guidance than Neural Diffusion Processes.
- The authors benchmark their method against existing approaches on both synthetic and real world data sets,  where  their method outperforms alternatives.

**Questions:**

- How did your results look without the added noise in the sample generation?
- Why are ERA5 results not reported with standard deviation?

**Ethical Concerns:**

["NO or VERY MINOR ethics concerns only"]

**Final Justification:**

The authors present a technically solid paper which addresses a signigificant shortcoming of prior work in neural processes. The research is well presented, and with the additional discussion and studies which they have indicated they will include for the camera ready version I am happy to recommend acceptance.

**Limitations:**

Yes.

**Paper Formatting Concerns:**

- Axes in Figure 2 should use bigger text for readability.
- Use the same number of significant figures in ERA5 as earlier experiments.

**Quality:**

3

**Strengths And Weaknesses:**

Strengths
- Well written paper with a good background for the field.
- Good benchmarking, building from simpler to more complex examples.
- NPs are an exciting field which show great promise in meta learning priors and performing inference and the proposed approach significantly improves the ability of the proposed method to make coherent predictions at large numbers of points as well as providing accurate predictions.
- While proposed method did not seem to require significant theoretical advances on previous methods, the two selected techniques compliment one another very well and solve a significant issue with Neural Processes.

Weaknesses
- The authors refer to their method as using an ODE solver for sampling, however they add noise in their sampling method, so their method more closely resembles sampling using Langevin dynamics. This addition to flow matching raises two concerns for me:
	- Empirically, I would have liked to see an ablation study comparing performance without this additional element.
	- Theoretically, this significantly changes how I would think about the proposed method, to something closer to an energy based model sampled with Langevin dynamics. I believe that this detail merited significantly more discussion in the main body of the paper.

---

> ### Author Rebuttal · Authors · 2025-07-30
>
> We thank the reviewers for their thorough review and constructive feedback, and we are happy that most reviewers recommend accepting our work.
>
> We address all the issues raised by the reviewers and as requested, include more detailed explanations and additional results that answer reviewers’ questions and further support our claims (see summary at the end of this response). We hope that the reviewers will acknowledge this.
>
> ## Deterministic vs. stochastic sample generation
>
> We agree that adding noise in the sampling process resembles Langevin dynamics. Though this can also be thought as a way to regularize the ODE solver, conceptually it is also equivalent to solving an SDE. We will add this discussion in the paper.
>
> Empirically, we add results that compare between stochastic and deterministic sampling as requested, both for 1D GP functions and for 2D images. The results for 1D are indistinguishable (as also noted in the paper). For images (specially for CelebA) we observe that the deterministic samples are a bit noisier while the stochastic samples are smoother.  When repeating this for the larger models trained for higher resolution CelebA (as requested by reviewer ZPkJ) this effect becomes even smaller and the deterministic and stochastic samples become very similar.
> This suggests that when the model is better, there is less need for stochasticity in sampling.
>
> We believe that adding this discussion and empirical results to the paper will improve its contribution and we thank the reviewer for pointing this out.
>
> ## ERA5 error bars
>
> We have now run trained our model for ERA5 on 5 different random seeds and obtain a mean log likelihood of 11.6647 with a standard deviation of 0.2452.
>
>
> # Summary of additional results (repeated in all responses):
> As requested, we have produced additional results to answer the reviewers’ questions and further support our claims. Some results include images that demonstrate qualitative aspects of samples, and will be added to the revised paper.
>
>
> We summarize the additional results:
>
> ### 1. Training with different flow schedules and computing likelihood with different ODE solvers.
>
> We have trained our model using various schedules for the RBF GP data. Each model is then evaluated on held out data by computing its log-likelihood. Our results are as follows (using the general schedule of $x_t = \alpha_t x_1 +  \beta_t x_0$ with different choices of $\alpha_t$ and $\beta_t$):
>
> | Name        | Schedule                                     | Log-Likelihood |
> |:------------|:---------------------------------------------|---------------:|
> | CondOT      | $\alpha_t = t, ~~ \beta_t = 1- t$            |         1.6890 |
> | Polynomial2 | $\alpha_t = t^2, ~~ \beta_t = (1- t)^2$     |         1.0955 |
> | Cosine      | $\alpha_t  = \sin(0.5 t \pi), ~~ \beta_t = \cos(0.5 t \pi)$            |         1.2441 |
> | LinearVP    | $\alpha_t = t, ~~ \beta_t = \sqrt{1- t^2}$            |         0.4112 |
>
>
> The results show schedules other than the standard CondOT perform worse in this setting.
>
> We also experiment with different ODE solvers for computing the log likelihood. For this we use our model trained with CondOT on RBF. We use a few different solvers using the implementation in the flow_matching package (based on torchdiffeq). The resulting  log likelihoods are:
>
> | ODE Solver | Log-Likelihood |
> |:-----------|---------------:|
> | Euler      |         1.7941 |
> | Midpoint   |         1.6890 |
> | RK4        |         1.6856 |
> | Dopri5     |         1.6857 |
>
> The results show that except for the simple Euler method, other methods result in very similar values.
> We will add both results above together with the same results on EMNIST to table 4 in the appendix to complete the ablation results.
>
> ### 2. Higher resolution CelebA samples
>
> We have trained our model on higher resolution CelebA images $64 \times 64$ and generated high quality conditional and unconditional samples. We will add these results to figure 6 in the appendix  (in the paper we showed results only for $32 \times 32$ resolution, following the experimental protocol of prior models).
>
>
> ### 3. Deterministic vs. stochastic sample generation
>
> We have added a qualitative comparison of deterministic sample generation (using an ODE solver), vs. a method that adds small amount of noise in the sampling process. As we mention in the paper, for the GP and ERA5 datasets we did not see any difference between the methods, however for image datasets (especially CelebA) the samples are a bit noisier in the deterministic case. This effect is even less visible in the larger model trained for resolution $64 \times 64$ (see additional results #2) where the deterministic and stochastic samples are very similar.
>
> ### 4. Error bars for ERA5
>
> We have repeated the model training for ERA5 data with 5 different seeds, and obtained a mean and std of 11.6647 +- 0.2452.

---

> > ### Comment · Reviewer_qfUz · 2025-08-02
> >
> > Thank you for addressing my concerns. With the addition of the material described in response to me and the other reviewers I am happy to increase my score.

---

### Official Review · Reviewer_MZUP · 2025-07-07

**Clarity:** 2
**Significance:** 3
**Originality:** 3
**Rating:** 5
**Confidence:** 2

**Summary:**

This paper proposes a probabilistic model called flow matching neural process that trains a neural process with the flow matching objective.
The main idea is to train a conditional flow matching model that maps from a Gaussian to the desired distribution at the target location.
They use a transformer architecture to parameterize the vector field so that the flow model is able to condition on arbitrary number of context tokens.

**Questions:**

1. As mentioned by the author, diffusion neural processes have already been published before.
Given the connection between diffusion models and flow matching models (in fact some would argue they are equivalent), I wonder if the method proposed in this paper apply to diffusion neural processes.
Also, the authors perhaps should elaborate a bit more on the key difference between this paper and diffusion neural processes.

**Ethical Concerns:**

["NO or VERY MINOR ethics concerns only"]

**Final Justification:**

I've read through the rebuttal and other reviews. I maintain my current score.

**Paper Formatting Concerns:**

No issues.

**Quality:**

3

**Strengths And Weaknesses:**

1. The idea seems to be very clean---combining transformers and flow matching objective for neural processes.
1. The empirical performance seems quite strong.
The proposed flow matching neural process outperforms all baselines consistently.

---

> ### Author Rebuttal · Authors · 2025-07-30
>
> We thank the reviewers for their thorough review and constructive feedback, and we are happy that most reviewers recommend accepting our work.
>
> We address all the issues raised by the reviewers and as requested, include more detailed explanations and additional results that answer reviewers’ questions and further support our claims (see summary at the end of this response). We hope that the reviewers will acknowledge this.
>
> ## FlowNP vs. Neural Diffusion Processes
>
> In the paper we describe the differences between FlowNP (our method) and Neural Diffusion Processes (NDP) in several places (e.g  lines 134-139 and in the appendix lines 465-471). In the revised version of the paper we will emphasize this in a clearer way.
> The differences can be summarized in the following three points:
> 1. FlowNP uses a linear conditional optimal transport noise schedule $x_t = t x_1 + (1-t) x_0$ and a model that predicts velocity, while NDP uses a variance preserving schedule and a model that predicts the noise. The ablations in table 4 directly compare this difference and we also add more noise schedules as requested by reviewer KAXg (see additional results #1 below).
> 2. FlowNP models joint and conditional distributions of points while NDP only models the joint distribution. The approach in NDP is to model the joint distributions of arbitrary number of points in the signal which allows computing also conditional distributions using Bayes' law. However, in order to sample from a conditional distribution, this requires adding auxiliary methods such as posterior sampling guidance. In our method, computing the conditional distribution is amortized during training and therefore the model can be directly used for probabilistic inference or sampling from any arbitrary conditional distribution.
> 3. While both FlowNP and NDP use transformers, the architectures are different. We explicitly leave this difference out of the paper's scope and in order to make a fair comparison we use the exact same architecture in both models (and in the TNP model) for all experiments.
>
> # Summary of additional results (repeated in all responses):
> As requested, we have produced additional results to answer the reviewers’ questions and further support our claims. Some results include images that demonstrate qualitative aspects of samples, and will be added to the revised paper.
>
>
> We summarize the additional results:
>
> ### 1. Training with different flow schedules and computing likelihood with different ODE solvers.
>
> We have trained our model using various schedules for the RBF GP data. Each model is then evaluated on held out data by computing its log-likelihood. Our results are as follows (using the general schedule of $x_t = \alpha_t x_1 +  \beta_t x_0$ with different choices of $\alpha_t$ and $\beta_t$):
>
> | Name        | Schedule                                     | Log-Likelihood |
> |:------------|:---------------------------------------------|---------------:|
> | CondOT      | $\alpha_t = t, ~~ \beta_t = 1- t$            |         1.6890 |
> | Polynomial2 | $\alpha_t = t^2, ~~ \beta_t = (1- t)^2$     |         1.0955 |
> | Cosine      | $\alpha_t  = \sin(0.5 t \pi), ~~ \beta_t = \cos(0.5 t \pi)$            |         1.2441 |
> | LinearVP    | $\alpha_t = t, ~~ \beta_t = \sqrt{1- t^2}$            |         0.4112 |
>
>
> The results show schedules other than the standard CondOT perform worse in this setting.
>
> We also experiment with different ODE solvers for computing the log likelihood. For this we use our model trained with CondOT on RBF. We use a few different solvers using the implementation in the flow_matching package (based on torchdiffeq). The resulting  log likelihoods are:
>
> | ODE Solver | Log-Likelihood |
> |:-----------|---------------:|
> | Euler      |         1.7941 |
> | Midpoint   |         1.6890 |
> | RK4        |         1.6856 |
> | Dopri5     |         1.6857 |
>
> The results show that except for the simple Euler method, other methods result in very similar values.
> We will add both results above together with the same results on EMNIST to table 4 in the appendix to complete the ablation results.
>
> ### 2. Higher resolution CelebA samples
>
> We have trained our model on higher resolution CelebA images $64 \times 64$ and generated high quality conditional and unconditional samples. We will add these results to figure 6 in the appendix  (in the paper we showed results only for $32 \times 32$ resolution, following the experimental protocol of prior models).
>
>
> ### 3. Deterministic vs. stochastic sample generation
>
> We have added a qualitative comparison of deterministic sample generation (using an ODE solver), vs. a method that adds small amount of noise in the sampling process. As we mention in the paper, for the GP and ERA5 datasets we did not see any difference between the methods, however for image datasets (especially CelebA) the samples are a bit noisier in the deterministic case. This effect is even less visible in the larger model trained for resolution $64 \times 64$ (see additional results #2) where the deterministic and stochastic samples are very similar.
>
> ### 4. Error bars for ERA5
>
> We have repeated the model training for ERA5 data with 5 different seeds, and obtained a mean and std of 11.6647 +- 0.2452.

---

### Note · Authors · 2025-08-14

We thank the AC and the reviewers for their effort.

Most reviewers initially recommended acceptance but there were some requests for clarifications and extra results.
As part of the rebuttal we have updated the paper and added more results as requested by the reviewers. All these results are summarized in the "Summary of extra results" section in our response.

We are happy that the reviewers found that our rebuttal addressed their remaining concerns and that most of them stated they had raised their score.

We will also incorporate the suggestion by reviewer ZPkJ to plot the images without interpolation.

---

### Decision · Program_Chairs · 2025-09-17

**Decision:**

Accept (poster)

**Comment:**

This paper introduces Flow Matching Neural Processes (FlowNP), which integrate flow matching into the Neural Process framework to enable efficient conditional sampling over continuous spaces. The method provides reasonable results across Gaussian processes, image inpainting, and weather data, with reviewers highlighting clarity after the rebuttal. While questions remain about scalability, positioning against non-NP baselines, and the extent of novelty, the paper is well presented justifying acceptance.